# Prevalence and risk factors of short birth interval in Bangladesh: Evidence from the linked data of population and health facility survey

**Mohammad Zahidul Islam[1,2]\*, M. Mofizul Islam[3], Md. Mostafizur Rahman[2], Md. Nuruzzaman Khan[1]**

**1** Department of Population Science, Jatiya Kabi Kazi Nazrul Islam University, Trishal, Mymensingh, Bangladesh, **2** Department of Population Science and Human Resource Development, Rajshahi University, Rajshahi, Bangladesh, **3** Department of Public Health, La Trobe University, Melbourne, Australia

\* zahid100779@gmail.com

**Data Availability Statement:** The datasets used and analysed in this study are available from the Measure DHS website: https://dhsprogram.com/data/available-datasets.cfm.

## Abstract

The Sustainable Development Goals 3 targets significant reductions in maternal and under-five deaths by 2030. The prevalence of these deaths is significantly associated with short birth intervals (SBI). Identification of factors associated with SBI is pivotal for intervening with appropriate programmes to reduce occurrence of SBI and associated adverse consequences. This study aimed to determine the factors associated with SBI in Bangladesh. A total of 5,941 women included in the 2017/18 Bangladesh Demographic and Health Survey 2017/18 and 1,524 healthcare facilities included in the 2017 Bangladesh Health Facility were linked and analysed. The sample was selected based on the availability of the birth interval data between the two most recent subsequent live birth. SBI was defined as an interval between consecutive births of 33 months or less, as recommended by the World Health Organization and was the outcome variable. Several individual-, households-, and community-level factors were considered as exposure variables. We used descriptive statistics to summarise respondents' characteristics and multilevel Poisson regression to assess the association between the outcome variable with exposure variables. Around 26% of live births occurred in short intervals, with a further higher prevalence among younger, uneducated, or rural women. The likelihoods of SBI were lower among women aged 20–34 years (PR, 0.14; 95% CI, 0.11–0.17) and ≥35 years (PR, 0.03; 95% CI, 0.02–0.05) as compared to the women aged 19 years or less. Women from households with the richest wealth quintile experienced lower odds of SBI (PR, 0.61; 95% CI, 0.45–0.85) compared to those from the poorest wealth quintile. The prevalences of SBI were higher among women whose second most recent child died (PR, 5.23; 95% CI, 4.18–6.55), those who were living in Chattogram (PR, 1.52; 95% CI, 1.12–2.07) or Sylhet (PR, 2.83, 95% CI, 2.08–3.86) divisions. Availability of modern contraceptives at the nearest healthcare facilities was 66% protective to the occurrence of SBI (PR, 0.34; 95% CI, 0.22–0.78). Also, the prevalence of SBI increased around 85% (PR, 1.85; 95% CI, 1.33–2.18) for every kilometer increase in the distance of nearby health facilities from women's homes. Targeted and tailored regional

**Funding:** The authors received no specific funding for this work.

**Competing interests:** The authors have declared that no competing interests exist.

policies and programmes are needed to increase the awareness of SBI and associated adverse health outcomes and availability of modern contraception in the healthcare facilities.

## Introduction

The World Health Organization (WHO) recommends at least 24 months interval between a live birth or termination to the next conception or 33 months interval between a live birth or termination to the next live birth [1]. The intervals shorter than these durations are considered as short birth intervals (SBI) which equates to around 25% of the live births globally with relatively high proportions in Central Asia (33%) and Sub-Saharan Africa (57%) [2]. SBI are found to be associated with several adverse health outcomes, including preterm birth [2, 3], low birth-weight [3, 4] and small for gestational age [4]. Neonatal and infant mortality are also found to be higher among children born in SBI [4, 5]. Maternal morbidities such as pre-eclampsia, high blood pressure, and premature rupture of the membrane are also found higher among mothers having a conception in short intervals of the previous birth. Importantly, these are the major causes of around 822 maternal deaths in a day in low- and lower-middle-income countries (LMICs) [6, 7].

In Bangladesh, around a quarter of pregnancies that end with live births are born in short intervals and the total number equates to over a million live births in a year. Like many other LMICs, this high number of births in short intervals caused several adverse health outcomes [8, 9], including maternal and child mortality [7] and prevented Bangladesh from achieving the relevant Millennium Development Goals of reducing maternal (target 5A: reduce maternal mortality to 143 per 100, 000 live births) and under-five mortality (target 4A: reduce under-five mortality to 50 per 1,000 live births) by 2015 [10]. Importantly, the Sustainable Development Goals for 2015–2030 also aim to achieve these targets, even though these are ambitious. For instance, targets 3.1 and 3.2 of the SDGs aim to reduce the maternal and under-five mortality rates to 70 (per 100,000 live births) and 25 (per 1,000 live births), respectively [11].

Improving maternal health and reducing the occurrences of adverse birth outcomes, including preterm birth and low birth weight, would be a key to achieving these targets in Bangladesh. A substantial reduction in SBI would be necessary through initiating relevant policies and programmes. Knowledge of the current prevalence of SBI and associated factors are essential for appropriate policy and programme development. However, there are at least two challenges in Bangladesh and other LMICs that make these difficult, which are (i) inaccurate measurement of SBI and (ii) inconsistent list of factors associated with SBI. Most studies conducted in Bangladesh and other similar settings used varied intervals from one birth to the next for measuring SBI, instead of the WHO recommendation of 33 months intervals. In the context of Bangladesh, previous studies used inconsistent duration in defining SBI, including <24 months [12], <33 months [13], <36 months [9] and <39 months [14].

The occurrence of SBI is mostly associated with the women's lack of understating about its adverse effects on their own health as well as their children's health and lower access to and use of family planning and contraception services [15–19]. Multidimensional factors affect access to healthcare services, including individual-, household-, community-, and health facility-level factors need to be studied to identify the key determinants of SBI [20]. Many of these factors are intertwined and need to be examined together. However, there is scarce literature particularly in the context of LMICs.

Previous studies on SBI considered only a limited set of confounding factors, mainly the individual-level factors such as women's age, age at marriage, number of children given birth,

nutritional status, educational achievements and economic conditions of the household, etc. [13, 17, 21, 22]. These factors are also intricately connected to inadequate access to health care services, particularly family planning and contraception services. Community-level norms such as having children soon after marriage, having several children to support family income in the future and/or safety reasons, desire for a child of a particular sex (particularly the male child) also play important determinants of SBI [14, 23–25].

Essentially, access to family planning and contraception services are influenced, to some extent, by health facility-level factors, including the availability and quality of family planning and contraception services near the women's residence [26]. This access can play an important role in reducing SBI by reducing the individual-level barriers particularly among disadvantaged women [27]. However, there exists little relevant research in this regard, mainly due to the lack of population- and health facility-level data. Although a few studies considered the distance of women's residence from the nearest healthcare facilities, they provide an incomplete understanding of the associations between health facility-level factors and SBI [13, 21]. Methodological challenges of the existing studies are also common, including small datasets [21] and ignoring the hierarchical structure of data where single-level analysis produces imprecise results [9, 12, 14, 28–30]. Using nationally representative population- and health facility-level data and advanced statistical modelling, this study examines the individual-, household-, community-, and health facility-level factors associated with SBI in Bangladesh.

## Methods

### Study overview

Data analysed in this study come from the nationally representative Bangladesh Demographic and Health Survey (BDHS) 2017/18 and Bangladesh Health Facility Survey (BHFS) 2017. The sampling procedure of both surveys can be found in the respective survey reports [31, 32].

The 2017/18 BDHS was conducted by the National Institute of Population Research and Training as part of the Demographic and Health Survey (DHS) programme and supervised by the Ministry of Health and Family Welfare of Bangladesh. The survey interviewed 20,127 reproductive-aged ever-married women selected through a two-stage stratified random sampling procedure. At the first stage, 675 primary sampling units (PSUs) were selected randomly from the list of 293,579 PSUs generated by the Bangladesh Bureau of Statistics as part of the 2011 National Population Census. Data were collected from 672 of these PSUs and the remaining three were excluded due to flood. At the second stage of sampling, 30 households were selected randomly from each of the selected PSUs. This produced a list of 20,160 households, among which interview was conducted in 19,457 households.

The 2017 BHFS is the source of healthcare facility data in Bangladesh. The survey collected data from 1,524 healthcare facilities selected proportionately from the public, private and non-government sectors. Both census and stratified random sampling methods were used to select these facilities from the 19,811 registered healthcare facilities in Bangladesh.

The GPS point locations are available for each of the 672 PSUs included in the 2017 BDHS and 1,524 healthcare facilities included in the 2017 BHFS. We linked the PSUs with the nearest healthcare facilities using the administrative boundary linkage method. The details of this method have been published elsewhere [33, 34].

### Sample

Of the 20,127 women included in the 2017/18 BDHS, 5,941 women were included in this study. The inclusion criteria were i) women had at least two pregnancies, of which the most recent one occurred within five years of the survey date and ended with live birth, (ii) the

second most recent pregnancy ended either with live birth or termination and (ii) the date of live birth after the most recent pregnancy and the date of birth or termination after the second most recent pregnancy were recorded.

## Outcome variable

In this study, SBI is the outcome variable. The BDHS recorded the interval between the end of the second last pregnancy through termination or live birth to the end of the most recent pregnancy through live birth. We categorised these data in line with the WHO's recommendation to define SBI. Thus, births were considered to have occurred in short intervals if the last live births occurred at 33 months or lesser intervals of the second last live births or terminations [1].

## Exposure variables

A comprehensive literature search was conducted in five databases (Medline, Embase, Web of Science, CINHAL, and Google Scholar) to identify potential determinants of SBIs [9, 12, 14–19, 30]. The factors identified and later found available in the dataset were considered. Later their associations with the SBIs were tested using the forward regression approach. Factors found significant at the p = 0.20 level were retained in the analysis. Finally, these factors were then summarised as individual-, household-, community-level factors following the socio-eco-logical model of health [35]. Individual-level factors were maternal age at birth ($\leq$19, 20–34, $\geq$35 years), maternal age at first birth (used as a continuous variable), maternal educational status (no formal education, primary, secondary, higher), and mothers' employment status at the time of the survey (employed, unemployed). Household-level factors were partners' educational status (no formal education, primary, secondary, higher), sex of the household's head (male, female), the total number of children ever given birth ($\leq$2, >2), exposure to mass media (little exposed, moderately exposed, highly exposed), survival status of the child born from the second most recent pregnancy (yes, no), and wealth quintile (poorest, poorer, middle, richer, richest). Place of residence (urban, rural) and region (seven administrative divisions, *namely* Barishal, Chattogram, Dhaka, Khulna, Mymensingh, Rajshahi, Rangpur, Sylhet) were the community-level factors. To our knowledge, this is the first study in Bangladesh and other LMICs that considered the health facility-level factors because of their potential role in modern contraception use in SBI [28–30, 36]. This is fourth level of socio-ecological model [35, 37]. We generated four health service variables: average distance to the nearest family planning facility, family planning service availability, readiness to provide family planning services, and general health facility readiness. The details of this approach can be found elsewhere [36, 38].

## Statistical analysis

Descriptive statistics with frequency and percentage distribution were used to describe the characteristics of the respondents. The distribution of SBI across exposure variables considered was also examined. Statistical significance of the difference in distribution was assessed using the chi-square test. Availability of modern contraceptives across facility types and division were also explored. The data were hierarchical, as respondents were nested in the households, and households nested in the PSU. Therefore, we expected responses from the same household and cluster would behave more alike than different households and clusters [39]. Moreover, the outcome of this study, i.e., SBI, is a common outcome. Previous studies found simple logistic regression models often overestimate the odds for a common outcome and hierarchical data [39, 40]. Therefore, we used multilevel Poisson regression model that accounts for these multiple hierarchies and dependency in data, and the problem of overestimation [41]. Five

models were run separately. Model 1 was the intercept-only model, presenting variance in prevalence of SBI across PSUs. Model 2 included all individual- and household-level factors. Model 3 included community-level factors. In model 4, health facility-level variables were included. Model 5 was the final model that included all factors. Multicollinearity was also examined. If evidence of multicollinearity was found (VIF>10), the relevant variable was deleted. Results were reported as Prevalence Ratios (PR) with 95% Confidence Intervals (95% CI). The Intra-Class Correlation (ICC), Variance Inflation Factor (VIF), AIC and BIC for each model were recorded and compared to select the best model. The ICC value was calculated by dividing the between-clusters-variance of SBI (random intercept variance) by the total variance of SBI (sum of between-clusters-variance and within-cluster (residual) variance of SBI). Geographical linkage between survey and health facility data was performed using the Statistical package R. All other statistical analyses were conducted using Stata version 15.1 (Stata Corp, College Station, Texas, USA).

**Ethics statement.** The BDHSs data collection methods and procedure were reviewed and approved by the institutional review board of ICF and the National Research Ethics Committee of the Bangladesh Medical Research Council. They also provided ethical approval of this survey. Informed consent was obtained from all participants during surveys and data were released in deidentified form. Therefore, no additional ethical approval was required for this study.

## Result

### Socio-demographic characteristics of the respondents

Table 1 shows the background characteristics of the respondents. A majority (86.24%) of women were 20–34 years old and one-fourth reported that they had their first birth before reaching 19. Around half of respondents had completed secondary-level education and 86% reported that the head of the household was a man. Overall, 8% of respondents reported that the child born from their second most recent pregnancy had died.

### Distribution of short birth intervals across socio-demographic characteristics

Over 26% of respondents had their last birth in a short interval of their second most recent birth (Table 1). The prevalence of SBI was higher among respondents aged 20–34 years (78%), who had their first birth ≤19 years (72.65%), or not engaged in formal work (57.23%). SBI was also higher among respondents who had ≤2 children at the time of the survey (56.56%), not exposed to mass media (41.51%) or had their child born from the second most pregnancy died (19.91%).

### Characteristics of health facilities and their distribution

The division-wise distribution of health facilities that provided modern contraceptives and their average distance from BDHS clusters are presented in Table 2.

Around 89% (1357/1524) of health facilities included in the 2017 BHFS provided modern contraceptives to the clients. Upazila (sub-unit of a district) to community-level government hospitals were the major providers of modern contraceptives (93%). The national average distance of modern contraception-providing health facilities from respondents' houses was 6.36 km. This distance was higher in the Sylhet division (8.34 km) and lower in the Chattogram division (5.85 km).

Table 1. Demographic characteristics of the respondents (n = 5,941).

| Characteristics | Total | Short birth intervals | | $p$ [1] |
|---|---|---|---|---|
| | | Yes (n = 1566) | No (n = 4375) | |
| **Maternal age at birth** | | | | |
| ≤19 | 463 (7.80) | 314 (20.06) | 149 (3.41) | <0.01 |
| 20–34 | 5124 (86.24) | 1222 (78.00) | 3902 (89.18) | |
| ≥35 | 354 (5.96) | 30 (1.93) | 324 (7.41) | |
| **Maternal age at first birth** | | | | |
| ≤19 | 4467 (75.19) | 1138 (72.65) | 3329 (76.10) | <0.05 |
| 20–34 | 1468 (24.71) | 424 (27.08) | 1044 (23.86) | |
| ≥35 | 6 (0.10) | 4 (0.27) | 2 (0.04) | |
| **Mothers' educational status** | | | | |
| No formal education | 568 (9.55) | 152 (9.66) | 416 (9.51) | <0.01 |
| Primary | 1961 (33.01) | 547 (34.95) | 1414 (32.31) | |
| Secondary | 2794 (47.03) | 681 (43.51) | 2113 (48.29) | |
| Higher | 618 (10.41) | 186 (11.87) | 432 (9.88) | |
| **Mothers' working status** | | | | |
| Yes | 2781 (46.81) | 670 (42.77) | 2111 (48.25) | <0.01 |
| No | 3160 (53.19) | 896 (57.23) | 2264 (51.75) | |
| **Husbands' educational status** | | | | |
| No formal education | 1079 (18.43) | 273 (17.72) | 806 (18.68) | <0.05 |
| Primary | 2179 (37.22) | 620 (40.15) | 1559 (36.17) | |
| Secondary | 1751 (29.91) | 412 (26.71) | 1339 (31.05) | |
| Higher | 846 (14.44) | 238 (15.42) | 608 (14.09) | |
| **Sex of the household's head** | | | | |
| Male | 5114 (86.09) | 1383 (88.34) | 3731 (85.28) | <0.05 |
| Female | 827 (13.91) | 183 (11.66) | 644 (14.72) | |
| **Total children ever born** | | | | |
| ≤2 | 3075 (51.75) | 886 (56.56) | 2189 (50.03) | <0.01 |
| >2 | 2866 (48.25) | 680 (43.44) | 2186 (49.97) | |
| **Exposure to mass media** | | | | |
| Little exposed | 2289 (38.52) | 650 (41.51) | 1639 (37.45) | <0.01 |
| Moderate exposed | 788 (13.26) | 250 (15.94) | 538 (12.31) | |
| Highly exposed | 2864 (48.21) | 666 (42.55) | 2198 (50.24) | |
| **Child born from the second most pregnancy was alive** | | | | |
| Yes | 5458 (91.88) | 1254 (80.09) | 4204 (96.10) | <0.01 |
| No | 483 (8.12) | 312 (19.91) | 171 (3.90) | |
| **Wealth quintile** | | | | |
| Poorest | 1439 (24.22) | 415 (26.51) | 1024 (23.40) | <0.01 |
| Poorer | 1287 (21.65) | 381 (24.30) | 906 (20.70) | |
| Middle | 1066 (17.95) | 260 (16.62) | 806 (18.43) | |
| Richer | 1097 (18.47) | 274 (17.50) | 823 (18.82) | |
| Richest | 1052 (17.71) | 236 (15.06) | 816 (18.66) | |
| **Type of residential place** | | | | |
| Urban | 1543 (25.97) | 375 (23.98) | 1168 (26.69) | <0.05 |
| Rural | 4398 (74.03) | 1191 (76.02) | 3207 (73.31) | |
| **Administrative division** | | | | |

*(Continued)*

**Table 1.** (Continued)

| Characteristics | Total | Short birth intervals | | $p$ [1] |
|---|---|---|---|---|
| | | Yes (n = 1566) | No (n = 4375) | |
| Barishal | 333 (5.61) | 71 (4.57) | 262 (5.99) | <0.01 |
| Chattogram | 1312 (22.08) | 379 (24.17) | 933 (21.33) | |
| Dhaka | 1466 (24.67) | 352 (22.48) | 1114 (25.45) | |
| Khulna | 502 (8.45) | 100 (6.41) | 402 (9.18) | |
| Mymensingh | 507 (8.54) | 142 (9.10) | 365 (8.34) | |
| Rajshahi | 643 (10.82) | 138 (8.78) | 505 (11.55) | |
| Rangpur | 626 (10.53) | 130 (8.27) | 496 (11.34) | |
| Sylhet | 552 (9.30) | 254 (16.22) | 298 (6.82) | |

Note

[1] The significance level was estimated using the Chi-square test. Fisher exact test was used for maternal age at first birth.

## Age-standardised prevalence of short birth intervals

We calculated the age-standardised prevalence of SBI across regions in Bangladesh (Fig 1, results not shown in the table) as there were substantial variations in the prevalences of SBI in Bangladesh across different age-groups of women. The distribution of women aged 15–49 in the 2011 national population census was used for standardisation. The age-standardised prevalence of SBI in the Sylhet region was 44.14%, almost two times higher than that in the Barishal region (22.20%).

## Model diagnosis

Of the five models conducted separately with individual-, household-, community-, and the health facility-level confounders, the best model was selected by comparing the model statistics including Akaike's information criterion (AIC), Bayesian information criterion (BIC), and intra-class correlation (ICC) (Table 3). The best model was the one that had the smallest AIC,

**Table 2. Distribution of health facilities that delivered modern contraception and their average distance from the clusters.**

| Administrative Division (%)* | Availability of modern contraceptives | | Type of health facilities where modern contraception was available | | | | | Average distance from facilities providing modern contraception (km) |
|---|---|---|---|---|---|---|---|---|
| | Yes (n = 1357) | No (n = 167) | District hospitals (n = 4) | Upazila (sub-district) and community-level government hospitals (n = 1271) | Mother and child welfare centres (n = 7) | NGO clinic or hospitals (n = 55) | Private hospitals (n = 21) | |
| Barishal (%) | 106 (93.7) | 7 (6.3) | 1(0.4) | 101 (95.9) | 1 (0.7) | 3 (2.7) | 0 (0.0) | 6.4 |
| Chattogram (%) | 261 (90.5) | 27 (9.5) | 1 (0.3) | 245 (93.9) | 1(0.5) | 9 (3.5) | 5 (1.8) | 5.9 |
| Dhaka (%) | 264 (87.1) | 39 (12.9) | 1 (0.3) | 234 (88.6) | 1(0.4) | 19 (7.3) | 9 (3.3) | 4.8 |
| Khulna (%) | 175 (93.2) | 13 (6.8) | 1 (0.3) | 165 (94.4) | 1(0.6) | 7 (4.2) | 1 (0.5) | 5.9 |
| Rajshahi (%) | 201 (91.5) | 19 (8.5) | 0 (0.0) | 191 (94.8) | 1(0.5) | 7 (3.3) | 3 (1.2) | 7.1 |
| Rangpur (%) | 147 (76.4) | 46 (23.6) | 0 (0.0) | 141 (95.6) | 1 (0.6) | 4 (2.2) | 2 (1.4) | 5.9 |
| Sylhet (%) | 93 (96.1) | 4 (3.9) | 0 (0.0) | 87 (94.2) | 1 (0.5) | 3 (3.5) | 1 (1.4) | 8.3 |
| Mymensingh (%) | 110 (90.0) | 12 (10.0) | 0 (0.0) | 106 (96.4) | 0 (0.0) | 3 (2.7) | 1 (0.9) | 6.44 |

Note: NGO—non-government organisation

*All are row percentages.

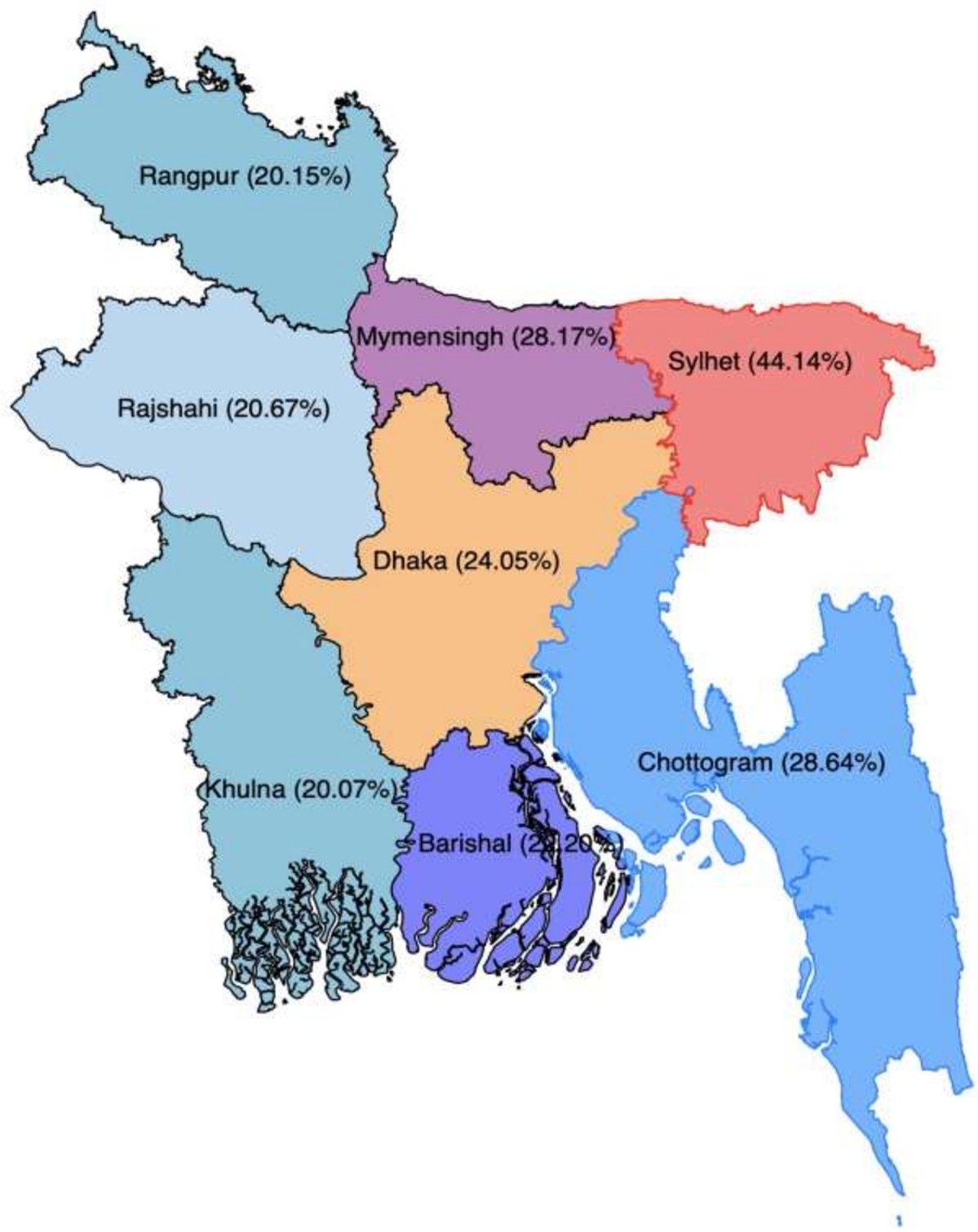

**Fig 1. Age-standardised prevalence of short birth intervals in Bangladesh, BDHS 2017/18.**

**Table 3. Results from random intercept model (measures of variation) to assess the factors associated with short birth intervals at the individual-, household-, community-, and health facility-level, BDHS 2017/18.**

| Random effects (Measures of variation for short birth intervals) | Model 1 | Model 2 | Model 3 | Model 4 | Model 5 |
|---|---|---|---|---|---|
| Community level variance (SE) | 0.85 (1.35)* | 1.20 (1.36)* | 0.46 (0.06)* | 0.37 (0.06)* | 0.42 (0.06)* |
| ICC | 0.19 | 0.13 | 0.06 | 0.05 | 0.04 |
| PCV | Reference | 27.78% | 66.67% | 72.22% | 77.78% |
| Median Odds Ratio | 2.25 | 3.15 | 1.54 | 1.84 | 1.49 |
| **Model fit statistics** | | | | | |
| AIC | 6980.60 | 6104.71 | 6756.93 | 7736.93 | 5927.44 |
| BIC | 7000.67 | 6251.57 | 6863.98 | 7864.98 | 6161.06 |

**Note:** Model 1 is the null model, a baseline model without any determinant variable. Model 2 is adjusted for individual- and household-level factors. Model 3 is adjusted for community-level factors. Model 4 is adjusted with health facility-level factors. Model 5 is the final model adjusted for the individual-, household- and community level-factors. AIC, Akaike's Information Criteria; BIC, Bayesian Information Criteria; PCV, Percentage Change in Variance.

*$p < 0.01$.

BIC and ICC. According to these markers, Model 5 that included individual-, household-, community-, and health facility-level variables was the best. The ICC in the null model was 18.92%, which was reduced to 4.12% once individual-, household-, community-, and health facility-level factors were adjusted in the Model 5.

## Predictors of short birth intervals

The results of the multilevel Poisson regression models examining factors associated with SBI are presented in Table 4. There was a decline in the prevalence of SBI with the increasing age of respondents. The PR was 86% (PR, 0.14; 95% CI, 0.11–0.17) and 97% (PR, 0.03; 95% CI, 0.02–0.05) lower among the respondents aged 20–34 years and ≥35 years, respectively, compared to the respondents aged ≤19 years. For each year increase in age at first birth leads to a 31% decrease in the prevalence of SBI (95% CI, 0.59–0.86). The PR was around 25% (PR,0.75; 95% CI, 0.61–0.93) lower among women residing in the female-headed households than those in the male-headed households. The PR of SBI was 5.23 (95% CI, 4.18–6.55) times higher for women whose second most recent child died compared to women whose second most recent child was alive. This study found a 39% (95% CI, 0.45–0.85) decline in the prevalence of SBI among respondents who belong to the richest wealth quintile than the respondents with the poorest wealth quintile. Respondents living in the Chattogram (PR, 1.52; 95% CI, 1.12–2.07) and Sylhet (PR, 2.83; 95% CI, 2.08–3.86) divisions reported a higher PR of SBI than those in the Barishal region. At the health facility level, we found PR of SBI declined around 32% (PR, 0.68; 95% CI, 0.54–0.95) and 66% (95% CI; 0.22–0.58) if the nearest health facilities provided the long-acting and short-acting modern contraceptives, respectively. We found that for every kilometer increase in distance of the nearest health facilities that provide modern contraceptives, women were 1.85 times (95% CI, 1.33–2.18) more likely to report a birth or pregnancy in short intervals.

## Discussion

Analysing the nationally representative data in a multilevel framework, in this study, we explored the prevalence of SBI in Bangladesh and factors associated with SBI. Factors found to be positively and significantly associated with SBI are lower maternal age at first birth, the experience of the death of the second most recent child, average distance of nearby health facilities from the respondents' cluster, and residential location in the Chattogram and Sylhet

**Table 4. Multilevel Poisson regression examining associated factors of short birth intervals in Bangladesh (n = 5,941).**

| Characteristics | Null model | Individual- and household-level model, PR (95% CI) | Community-level model, PR (95% CI) | Health facility model, PR (95% CI) | Individual-, household-, community-, health facility-level model, PR (95% CI) |
|---|---|---|---|---|---|
| **Maternal age at birth** | | | | | |
| ≤19 (ref) | | 1.00 | | | 1.00 |
| 20–34 | | 0.09 (0.01–0.63)* | | | 0.14 (0.11–0.17)** |
| ≥35 | | 0.02 (0.001–0.37)* | | | 0.03 (0.02–0.05)** |
| **Maternal age at first birth** | | 0.71 (0.56–0.89)* | | | 0.69 (0.59–0.86)** |
| **Mothers' educational status** | | | | | |
| No formal education (ref) | | 1.00 | | | 1.00 |
| Primary | | 0.78 (0.55–1.12) | | | 0.88 (0.68–1.14) |
| Secondary | | 0.73 (0.48–1.09) | | | 0.92 (0.70–1.21) |
| Higher | | 1.08 (0.71–1.65) | | | 1.39 (0.96–2.02) |
| **Mothers' employment status** | | | | | |
| No (ref) | | 1.00 | | | 1.00 |
| Yes | | 0.64 (0.44–0.94)* | | | 0.87 (0.75–1.01) |
| **Husbands' educational status** | | | | | |
| No formal education (ref) | | 1.00 | | | 1.00 |
| Primary | | 1.19 (0.92–1.55) | | | 1.17 (0.96–1.43) |
| Secondary | | 1.00 (0.78–1.30) | | | 1.02 (0.81–1.27) |
| Higher | | 1.15 (0.80–1.65) | | | 1.23 (0.92–1.65) |
| **Sex of the household head** | | | | | |
| Male (ref) | | 1.00 | | | 1.00 |
| Female | | 0.72 (0.51–1.02) | | | 0.75 (0.61–0.93)** |
| **Total children ever born** | | | | | |
| ≤2 (ref) | | 1.00 | | | 1.00 |
| >2 | | 1.21 (0.96–1.52) | | | 1.05 (0.91–1.22) |
| **Exposure to mass media** | | | | | |
| Little exposed (ref) | | 1.00 | | | 1.00 |
| Moderate exposed | | 1.13 (0.86–1.47) | | | 1.22 (0.98–1.51) |
| Highly exposed | | 0.75 (0.57–1.00)* | | | 0.92 (0.77–1.09) |
| **Sibling's survival status** | | | | | |
| Yes (ref) | | 1.00 | | | 1.00 |
| No | | 7.14 (1.49–34.24)* | | | 5.23 (4.18–6.55)** |
| **Wealth quintile** | | | | | |
| Poorest (ref) | | 1.00 | | | 1.00 |
| Poorer | | 1.18 (0.91–1.54) | | | 1.12 (0.91–1.37) |
| Middle | | 0.93 (0.70–1.22) | | | 0.93 (0.73–1.19) |
| Richer | | 0.91 (0.68–1.20) | | | 0.84 (0.65–1.09) |
| Richest | | 0.62 (0.38–1.01) | | | 0.61 (0.45–0.85)** |
| **Type of residential place** | | | | | |
| Urban (ref) | | | 1.00 | | 1.00 |
| Rural | | | 0.95 (0.79–1.16) | | 0.96 (0.79–1.17) |
| **Division** | | | | | |
| Barishal (ref) | | | 1.00 | | 1.00 |
| Chattogram | | | 1.40 (1.05–1.87)* | | 1.52 (1.12–2.07)** |
| Dhaka | | | 1.23 (0.90–1.68) | | 1.25 (0.90–1.74) |
| Khulna | | | 0.95 (0.68–1.32) | | 0.99 (0.69–1.41) |
| Mymensingh | | | 1.34 (0.98–1.81) | | 1.34 (0.97–1.85) |

*(Continued)*

**Table 4.** (Continued)

| Characteristics | Null model | Individual- and household-level model, PR (95% CI) | Community-level model, PR (95% CI) | Health facility model, PR (95% CI) | Individual-, household-, community-, health facility-level model, PR (95% CI) |
|---|---|---|---|---|---|
| Rajshahi | | | 0.92 (0.66–1.28) | | 0.97 (0.69–1.38) |
| Rangpur | | | 0.85 (0.62–1.17) | | 0.83 (0.59–1.16) |
| Sylhet | | | 2.72 (2.03–3.64)** | | 2.83 (2.08–3.86)** |
| **General health service readiness scores** | | | | | |
| Health facility management system | | | | 0.92 (0.88–1.19) | 0.85 (0.75–1.14) |
| Health facility infrastructure | | | | 0.86 (0.74–1.12) | 0.78 (0.64–1.10) |
| **Scores of availability of modern contraceptives type** | | | | | |
| Long-acting contraceptives | | | | 0.78 (0.64–0.95) | 0.68 (0.54–0.95)** |
| Short-acting contraceptives | | | | 0.45 (0.32–0.98) | 0.34 (0.22–0.78)** |
| **Facility's readiness scores** | | | | | |
| Long-acting contraceptives | | | | 0.95 (0.48–1.13) | 1.16 (0.96–1.46) |
| Short term contraceptives | | | | 0.67 (0.54–1.17) | 1.45 (0.98–1.48) |
| **Average distance to the nearest health facility** | | | | 1.45 (1.23–1.75) | 1.85 (1.33–2.18)** |

Note

*p<0.05

**p<0.01. Model summary for each model is presented in Table 3.

divisions. The availability of modern contraceptives at the nearest health facilities and female-headed households were found to be protective of SBI.

The prevalence of SBI found in this study (26%) is slightly higher than the global average (25%) [42], but it is in the global range (19.4% to 65.9%) [43, 44]. However, these global estimates used mostly 24 months interval to compute SBI, whereas, in this study, we considered 33 months following the WHO guideline. In that consideration, our observed prevalence may be lower than a global estimate of SBI based on the WHO guideline. This estimate for Bangladesh, which may be relevant for other LMICs, will help make evidence-based policies and programmes for SBI reduction. Indeed, the target for SBI reduction was rooted around two decades ago, in 2000, as part of the Millennium Development Goals and remains a priority in the Sustainable Development Goals [9, 12–14, 30]. The focus of the current development goals is to improve maternal health and reduce adverse consequences during pregnancy and childbirth, which are major causes of maternal and child mortality in LMICs [45, 46].

We found a higher likelihood of SBI among women who gave their most recent birth at the age of 19 or earlier. Our finding of a relatively high prevalence of SBI among younger women is similar to that reported in some studies conducted in LMICs [17, 42, 47, 48]. In most LMICs, many women marry before reaching 18. Younger women are sexually more active than older women. Also, younger women are generally more fertile [49], and at this early age, they are unaware of reproductive health, contraception, and the importance of birth spacing. Besides, the social norm plays a key role both in early marriage and service-seeking behaviour after pregnancy. For instance, many women avoid facility-based reproductive healthcare services, as they feel uncomfortable in seeking services from male providers and there is a shortage of female providers [50]. Moreover, the literature suggests that very young women are

likely to be economically disadvantaged, which may further restrict their access to and use of modern contraception and possibly explains this association [51, 52].

This study found a lower likelihood of SBI among participants living in households headed by a woman. In our data, around 14% of households were headed by a woman. Although no study so far has explored the reasons for such association, there may have several explanations. First, women are usually more aware than men about pregnancy complication and their connection with SBI. They may have gained this awareness from their fertility experiences. If they are the heads, they may avoid SBI for themselves and/or recommend others in their households to avoid shorter intervals. Second, in some of these households, the study participants may have identified themselves as the heads because their husbands were living abroad. In such circumstances, they have a lower probability of having pregnancies in short intervals. Third, if women identified themselves as the head of the households, then they may have been enjoying decision-making power about birth intervals and/or the use of contraception.

Consistent with studies conducted in some other LMICs [16, 53], this study found that the well-off women were less likely to give birth in short intervals than the least well-off. Usually, less frequent [54, 55] and less effective contraception use [54, 55], no knowledge of emergency contraception [54], or the combinations of these contraception-using behaviours are more common among the less affluent women. These behaviours lead to a consistently higher risk of unintended pregnancy [50] and most of them in short intervals. Also, women's knowledge and affordability of modern contraception use are linked to their economic status. Moreover, less affluent women usually have lower autonomy over their reproductive life and access to health facilities. Women of poor households are subject to religious and social norms that may restrict their movements in a male-dominated society. Indeed, several social determinants of health are interlinked, and they both separately and together may affect women's access to and use of modern contraception.

We found women's experience of the death of a child born from their second recent pregnancy was a strong predictor of SBI. This association is consistent with previous findings of some studies conducted in LMICs [16, 56–58] and there are at least two possible explanations for this. First, the death of an infant suddenly terminates breastfeeding [59–61]. This, in turn, triggers the resumption of ovulation and thereby increases the exposure duration of a new conception. Second, a previous adverse outcome may have a "replacement effect" [62, 63], which prompts women to soon get pregnant in an attempt to attain the desired number of surviving offspring at the end of their reproductive life. Third, depression from the experience of a child's death may also make women unaware of or indifferent to contraception use.

Provision of modern contraceptives at the nearest health facility from women's houses is a strong negative predictor of SBI. Also, the prevalence of SBI increases with an increasing average distance of the nearest health facilities from women's residences. These relationships are reasonable, as the provision increases the availability and the smaller distance increases the accessibility [20]. Service convenience is crucial particularly in the post-partum period when movement is considered risky. Women in Bangladesh usually face restrictions in travelling long distances and these restrictions are linked with gender, community norms, transport and time barriers.

To our knowledge, this is the first study in the context of LMICs that examines the determinants of SBI using the linked population survey and health facility data. The linking of two national datasets enabled us to adjust the influence of health facility-level factors, including the availability of modern contraceptives and the average distance of women's houses in the respective clusters from health facilities. These facility-level factors substantially determine the prevalence of SBI and often mediate the influence of individual-, household-, and community-level factors of SBI. Moreover, we used a robust statistical model-building technique and

adjusted for individual-, household-, community-, and health facility-level confounders, which ensured the precision of our findings. Thus, we believe, our findings will inform evidence-based policies and programmes in Bangladesh and other LMICs in reducing the prevalence of SBI. However, a limitation of this study is its cross-sectional data; therefore, the associations are correlational rather than causal. To ensure the privacy of the respondents, the DHS displaced BDHS cluster locations up to five km in rural areas and two km in urban areas. This displacement may have changed the average distance of the nearest health facilities from women's houses and other estimates presented in this study. However, the DHS ensured that the clusters are located in their original administrative division. Although this displacement may have changed the average distance of the nearest health facilities from women's houses and other estimates presented in this study, this variation is likely to be random. Also, some women may have sourced contraceptives from health facilities located further away than closest ones and/or from private facilities.

## Conclusion

This study found more than one-fourth of live births in Bangladesh occurred in short intervals. The prevalence of SBI was lower among women of relatively high ages, residing in the households headed by a female, and of the richest wealth quintile. The availability of modern contraceptives at the nearest health facility from women's respective clusters is protective of SBI. Also, women's experience of the death of their second most recent child is a strong predictor of SBI. Increased availability of modern contraceptives in all health facilities and targeted programmes for women of younger ages or women who had adverse pregnancy experiences should be prioritized in the policies and programmes to reduce SBI and associated adverse outcomes. Further studies to identify the geographical hotspots and associated factors of SBI may also help in targeted interventions.

## Author Contributions

**Conceptualization:** Mohammad Zahidul Islam.

**Formal analysis:** Mohammad Zahidul Islam.

**Methodology:** Mohammad Zahidul Islam.

**Project administration:** Mohammad Zahidul Islam.

**Software:** Mohammad Zahidul Islam.

**Supervision:** Md. Mostafizur Rahman, Md. Nuruzzaman Khan.

**Validation:** Mohammad Zahidul Islam.

**Writing – original draft:** Mohammad Zahidul Islam.

**Writing – review & editing:** M. Mofizul Islam, Md. Mostafizur Rahman, Md. Nuruzzaman Khan.

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
