## [Decision Letter · Decision Letter 0]

22 Dec 2021

PGPH-D-21-00677

Prevalence and risk factors of short birth interval in Bangladesh: Evidence from the linked data of population and health facility survey

Dear Dr.Mohammad Zahidul Islam ,

Thank you for submitting your manuscript to PLOS Global Public Health. After careful consideration, we feel that it has merit but does not fully meet PLOS Global Public Health’s publication criteria as it currently stands. Therefore, we invite you to submit a revised version of the manuscript that addresses the points raised during the review process.

We look forward to receiving your revised manuscript.

Kind regards,

Lungiswa Nkonki, PhD

Academic Editor

Journal Requirements:

1. Please provide separate figure files in .tif or .eps format only.  Please ensure that all files are under our size limit of 20MB.  

For more information about how to convert your figure files please see our guidelines: Once you've converted your files to .tif or .eps, please also make sure that your figures meet our format requirements

2. Please update the completed 'Competing Interests' statement, including any COIs declared by your co-authors. If you have no competing interests to declare, please state "The authors have declared that no competing interests exist".

3. We see that your study includes live participants, but you have not included an Ethics Statement. Please update your manuscript file to include an Ethics Statement subsection to your Materials and Methods section. It should include:

iii) (for human participants or donors) - A statement that formal consent was obtained (must state whether verbal/written) OR the reason consent was not obtained (e.g. anonymity)

Additional Editor Comments (if provided):

Reviewer 1

Thank you to the authors for the opportunity to read and review the article. Some comments for consideration: The major contribution from this paper is really that living closer to health facilities is a strong protective predictor of SBI. To strengthen this aspect of the paper, I would restructure the paper to make this the focus of the analysis. This would include writing more about this point in the introduction, discussing what other studies have found on the matter and perhaps doing more decomposed analysis focusing on this. It is also key that this be structured within the existing theory of health care access. Perhaps the authors could refer to Levesque et al.'s theory on access to healthcare: https://link.springer.com/article/10.1186/1475-9276-12-18/ This discussion could include a stipulation of the potential transmission mechanisms as well. The authors state that the analysis is cross-sectional, but with the number of observations, I would recommend they use a matching technique to say something causal about the relationship. Page 4: "Like many other LMICs, this high number of births in short intervals caused a large-scale of several adverse health outcomes". There seems to be a word missing after large-scale. Page 4 "which will require substantially reduce the prevalence of births in SBI through initiating policies and programmes". Should this be 'require a substantial reduction'? The research question needs to be more clearly stated in the end of the Introduction section. The authors seem to be using different fonts on page 6. It is unclear to me how the two datasets will be linked. Are you able to link individuals from the two data sets to one another? I realised later that you are linking individuals to data on their closest health facility, but this should be made clear in the data section. What is a Upazila? Page 6: "The inclusion criteria were i) had at least two pregnancies" - there seems to be a word missing here. Page 7 "survival status of the child born from the second most pregnancy" - should this be 'most recent'? Page 12 "but it is in the range of global range" - perhaps do not use range twice. Page 13 "they have a lower probability of getting pregnancies in short intervals." - should this be "having pregnancies"?

Reviewer 2

Review report submitted to PLOS Global Public Health Manuscript Title: Prevalence and risk factors of short birth interval in Bangladesh: Evidence from the linked data of population and health facility survey Manuscript ID: PGPH-D-21-00677 Corresponding author: Mohammad Zahidul Islam Summary The authors assessed the predictors of short birth interval in Bangladesh by linking two national databases. Using descriptive and multilevel analysis, the study found that individual, household, and community level factors, and availability of modern contraceptives at health facility were associated with short birth interval. This study is relevant given that short birth interval remains a leading cause of maternal and child mortality in developing countries. A strength of this study is the nationally representative data. However, the authors did not provide literature to support healthcare facility-level factors and short birth interval in the Introduction. Moreover, the selection of potential predictors was not based on adequate evidence from systematic reviews. I suggest the authors rewrite the Introduction to include the relevance of healthcare facility level factors and short birth interval. The language is unclear. I advise that the authors revise the language to improve the readability. Below are specific comments to improve the manuscript. Major revision Introduction 1. The authors should rewrite this section to integrate the relevant literature on predictors of short birth interval, healthcare facility-level factors, and short birth interval. Methods 2. The list of potential determinants was not comprehensive. Other factors e.g., breastfeeding, the sex of previous child have been found to be the only significant factors in a recent systematic review published by Pimentel et al. BMC Pregnancy and Childbirth 2020. The paper could be strengthened by including these factors in the models. Discussion 3. How does the findings on healthcare facility relate to the literature? Minor revision Abstract 4. The authors stated “The Sustainable Development Goals target significant reductions…..” The authors should be specific by stating which goal(s). 5. They should present additional information on the sample (respondents), the inclusion criteria, and sample size. 6. Provide statistics with 95% CI on education and rural women 7. “The prevalences of SBI were higher among women for whom the children born from the second most pregnancies died” This statement is unclear. Please revise. 8. In the conclusion, the authors should be specific and indicate who the targets of interventions should be, based on the findings from the study. Introduction 9. State the specific goals and objective that target maternal and child mortality. 10. The authors provided the prevalence of SBI as 26% in the introduction. How different is this statistic from what has been presented in the results? Methods 11. Ethics statement is missing in the text. I advise they include statement on ethics consideration as well as describe the process of obtaining permission to access the data. 12. Statistical analysis: Could the authors describe how they handled missing data in the analysis? 13. How were the exposure variables selected? 14. What level of significance was considered for an exposure variable to be retained in the analysis? Results 15. “Distribution of short birth intervals across socio-demographic characteristics” indicate where to find this in the manuscript 16. The prevalence of short birth interval had been reported in the Introduction. It is not clear why the authors included existing statistics in the results. They should provide information to justify the decision. Discussion 17. This contradicts the information in the abstract where richest wealth quintile had lower odds of SBI. This should be revised.

Reviewers' comments:

Reviewer's Responses to Questions

**Comments to the Author**

1. Does this manuscript meet PLOS Global Public Health’s publication criteria? Is the manuscript technically sound, and do the data support the conclusions? The manuscript must describe methodologically and ethically rigorous research with conclusions that are appropriately drawn based on the data presented.

Reviewer #1: Partly

Reviewer #2: Partly

2. Has the statistical analysis been performed appropriately and rigorously?

Reviewer #1: Yes

Reviewer #2: I don't know

3. Have the authors made all data underlying the findings in their manuscript fully available (please refer to the Data Availability Statement at the start of the manuscript PDF file)?

Reviewer #1: Yes

Reviewer #2: Yes

4. Is the manuscript presented in an intelligible fashion and written in standard English?

Reviewer #1: No

Reviewer #2: Yes

5. Review Comments to the Author

Reviewer #1: Thank you to the authors for the opportunity to read and review the article. Some comments for consideration:

The major contribution from this paper is really that living closer to health facilities is a strong protective predictor of SBI. To strengthen this aspect of the paper, I would restructure the paper to make this the focus of the analysis. This would include writing more about this point in the introduction, discussing what other studies have found on the matter and perhaps doing more decomposed analysis focusing on this. 

It is also key that this be structured within the existing theory of health care access. Perhaps the authors could refer to Levesque et al.'s theory on access to healthcare: https://link.springer.com/article/10.1186/1475-9276-12-18/ This discussion could include a stipulation of the potential transmission mechanisms as well. 

The authors state that the analysis is cross-sectional, but with the number of observations, I would recommend they use a matching technique to say something causal about the relationship. 

Page 4:  "Like many other LMICs, this high number of births in short intervals caused a large-scale of several adverse health outcomes". There seems to be a word missing after large-scale. 

Page 4 "which will require substantially reduce the prevalence of births in SBI through  initiating policies and programmes". Should this be 'require a substantial reduction'?

The research question needs to be more clearly stated in the end of the Introduction section.  

The authors seem to be using different fonts on page 6.

It is unclear to me how the two datasets will be linked. Are you able to link individuals from the two data sets to one another? I realised later that you are linking individuals to data on their closest health facility, but this should be made clear in the data section. 

What is a Upazila?

Page 6: "The inclusion criteria were i) had at least two pregnancies" - there seems to be a word missing here. 

Page 7 "survival status of the child born from the second most pregnancy" - should this be 'most recent'?

Page 12 "but it is in the range of global range" - perhaps do not use range twice. 

Page 13 "they have a lower probability of getting pregnancies inshort intervals." - should this be "having pregnancies"?

Reviewer #2: Review report submitted to PLOS Global Public Health

Manuscript Title: Prevalence and risk factors of short birth interval in Bangladesh: Evidence from the linked data of population and health facility survey

Manuscript ID: PGPH-D-21-00677

Corresponding author: Mohammad Zahidul Islam

Summary

The authors assessed the predictors of short birth interval in Bangladesh by linking two national databases. Using descriptive and multilevel analysis, the study found that individual, household, and community level factors, and availability of modern contraceptives at health facility were associated with short birth interval. This study is relevant given that short birth interval remains a leading cause of maternal and child mortality in developing countries. A strength of this study is the nationally representative data. However, the authors did not provide literature to support healthcare facility-level factors and short birth interval in the Introduction. Moreover, the selection of potential predictors was not based on adequate evidence from systematic reviews. I suggest the authors rewrite the Introduction to include the relevance of healthcare facility level factors and short birth interval. The language is unclear. I advise that the authors revise the language to improve the readability.

Below are specific comments to improve the manuscript.

Major revision

Introduction

1. The authors should rewrite this section to integrate the relevant literature on predictors of short birth interval, healthcare facility-level factors, and short birth interval.

Methods

2. The list of potential determinants was not comprehensive. Other factors e.g., breastfeeding, the sex of previous child have been found to be the only significant factors in a recent systematic review published by Pimentel et al. BMC Pregnancy and Childbirth 2020. The paper could be strengthened by including these factors in the models.

Discussion

3. How does the findings on healthcare facility relate to the literature?

Minor revision

Abstract

4. The authors stated “The Sustainable Development Goals target significant reductions…..” The authors should be specific by stating which goal(s).

5. They should present additional information on the sample (respondents), the inclusion criteria, and sample size.

6. Provide statistics with 95% CI on education and rural women

7. “The prevalences of SBI were higher among women for whom the children born from the second most pregnancies died” This statement is unclear. Please revise.

8. In the conclusion, the authors should be specific and indicate who the targets of interventions should be, based on the findings from the study.

Introduction

9. State the specific goals and objective that target maternal and child mortality.

10. The authors provided the prevalence of SBI as 26% in the introduction. How different is this statistic from what has been presented in the results?

Methods

11. Ethics statement is missing in the text. I advise they include statement on ethics consideration as well as describe the process of obtaining permission to access the data.

12. Statistical analysis: Could the authors describe how they handled missing data in the analysis?

13. How were the exposure variables selected?

14. What level of significance was considered for an exposure variable to be retained in the analysis?

Results

15. “Distribution of short birth intervals across socio-demographic characteristics” indicate where to find this in the manuscript

16. The prevalence of short birth interval had been reported in the Introduction. It is not clear why the authors included existing statistics in the results. They should provide information to justify the decision.

Discussion

17. This contradicts the information in the abstract where richest wealth quintile had lower odds of SBI. This should be revised.

6. PLOS authors have the option to publish the peer review history of their article (what does this mean?). If published, this will include your full peer review and any attached files.

**Do you want your identity to be public for this peer review?** For information about this choice, including consent withdrawal, please see our Privacy Policy.

Reviewer #1: No

Reviewer #2: No

---

## [Editor Report · Decision Letter 1]

2 Mar 2022

Prevalence and risk factors of short birth interval in Bangladesh: Evidence from the linked data of population and health facility survey

PGPH-D-21-00677R1

Dear Mr. Islam,

We are pleased to inform you that your manuscript 'Prevalence and risk factors of short birth interval in Bangladesh: Evidence from the linked data of population and health facility survey' has been provisionally accepted for publication in PLOS Global Public Health.

Best regards,

Lungiswa Nkonki, PhD

Academic Editor